# Using OPLS-DA to Fingerprint Key Free Amino and Fatty Acids in Understanding the Influence of High Pressure Processing in New Zealand Clams

**DOI:** 10.3390/foods12061162

**Published:** 2023-03-09

**Authors:** Muhammad Syahmeer How, Nazimah Hamid, Ye Liu, Kevin Kantono, Indrawati Oey, Mingfei Wang

**Affiliations:** 1Department of Process and Food Engineering, Faculty of Engineering, Universiti Putra Malaysia, Serdang 43400, Selangor Darul Ehsan, Malaysia; 2Department of Food Science, Auckland University of Technology, Private Bag 92006, Auckland 1142, New Zealand; 3Department of Food Science, University of Otago, P.O. Box 56, Dunedin 9054, New Zealand; 4Riddet Institute, Private Bag 11222, Palmerston North 4442, New Zealand

**Keywords:** high pressure processing, amino acid, fatty acid, OPLSDA

## Abstract

This study investigated the effect of high pressure processing (HPP) on the fatty acids and amino acids content in New Zealand Diamond Shell (*Spisula aequilatera*), Storm Shell (*Mactra murchisoni*), and Tua Tua (*Paphies donacina*) clams. The clam samples were subjected to HPP with varying levels of pressure (100, 200, 300, 400, 500, and 600 MPa) and holding times (5 and 600 s) at 20 °C. Partial Least Squares Discriminant Analysis (PLS-DA) and Orthogonal Partial Least Squares Discriminant Analysis (OPLS-DA) were deployed to fingerprint the discriminating amino and fatty acids post-HPP processing while considering their inherent biological variation. Aspartic acid (ASP), isoleucine (ILE), leucine (LEU), lysine (LYS), methionine (MET), serine (SER), threonine (THR), and valine (VAL) were identified as discriminating amino acids, while C18:0, C22:1n9, C24:0, and C25:5n3 were identified as discriminating fatty acids. These amino and fatty acids were then subjected to mixed model ANOVA. Mixed model ANOVA was employed to investigate the influence of HPP pressure and holding times on amino acids and fatty acids in New Zealand clams. A significant effect of pressure levels was reported for all three clam species for both amino and fatty acids composition. Additionally, holding time was a significant factor that mainly influenced amino acid content. butnot fatty acids, suggesting that hydrostatic pressure hardly causes hydrolysis of triglycerides. This study demonstrates the applicability of OPLS-DA in identifying the key discriminating chemical components prior to traditional ANOVA analysis. Results from this research indicate that lower pressure and shorter holding time (100 MPa and 5 s) resulted in the least changes in amino and fatty acids content of clams.

## 1. Introduction

New Zealand clams generally yield between 28 to 35% percentage flesh to shell, in the same range as clams often found on the coasts of the U.S and Canada, such as *Mya arenaria* (29%) and lower than Manila clam (*Venerupis philippinarum*) (44%) [1,2]. Due to its mild, sweet ocean flavor and crisp texture, clam meat is often consumed either raw or cooked. To avoid rancidity and extend shelf life, the common way of processing clam for export is either by chilling or freezing. Although freezing can extend the shelf life of seafood better than chilling, it can cause some undesirable effects. For example, it was found that the polyunsaturated and monounsaturated fatty acid contents of Tunisian mussel (*Mytilus galloprovincialis*) significantly decreased after 10 and 15 days of frozen storage at −20 °C [3]. In addition, freezing at −18 °C for three months significantly reduced the total amount of essential and non-essential amino acids in Nile Tilapia fish [4]. Hence, in this study, it was of interest to find an alternative method that can combine the advantages of both chilling and freezing methods.

The application of HPP treatment in the food industry has increased in the past few decades [5]. HPP is a cold pasteurization technique that can eliminate pathogens, extend shelf life, improve texture (subsequently influencing oral food processing and gastrointestinal digestibility), and maintain food quality [6,7,8,9,10,11]. In HPP, the high level of isostatic pressure is transmitted by water to a sealed food product, which is placed in a special container [12]. There are a number of indicators that can be examined to determine the effectiveness of HPP on meat quality, such as tenderness, juiciness, appearance (color and structure), fat and protein content, drip and cooking loss, fat quality and off-odors [13]. 

Fatty acids (FAs) content is a particularly important quality indicator in HPP processed meat. Thermal treatment can result in lipid oxidation, which is a major cause of meat quality deterioration due to rancidity and formation of off-flavors [14]. FAs can also reduce the risk of coronary heart disease and cancer, as well as improve inflammatory conditions [15]. Previous studies investigating the effect of HPP treatment on FAs content have mostly focused on poultry products, and studies on seafood are somewhat limited [16]. FA values are commonly used as a measure of rancidity that is expressed as oleic acid equivalent (g/100 g sample). The influence of high pressure treatment on FAs varied with different types of fish [17,18,19,20]. Higher pressure holding time and longer storage time favor the formation of FAs [17,18,21]. In oysters, only one monounsaturated fatty acid, C18:1, was found to significantly decrease after pressure treatment [22,23].

Free amino acids (FAAs) content is another important indicator to evaluate the effectiveness of HPP on the quality of meat. FAAs are susceptible to oxidative reactions during thermal treatments, and can result in a decrease in protein digestibility, which in turn affects the nutritional value of meat [14]. In addition, the color and texture deterioration of meat has been related to protein oxidation [24]. The presence of amino acids and peptides have also been linked to the development of meat flavor [25]. In terms of seafood, only one study has investigated the effect of HPP treatments on the amino acid content of squid at 200, 400, and 600 MPa for 10 min [26]. The authors reported no significant differences in essential amino acid content. In addition, pressure levels at 200 and 400 MPa were found to significantly increase a few non-essential amino acids (proline, glycine, and tyrosine).

To the authors’ knowledge, no studies have been carried out to investigate the influence of HPP on both the amino and fatty acids content of clams during chilled storage. Prior studies on clams have only examined the effect of HPP on their microbiological quality, and properties such as surface area and moisture content as quality indicators [27,28,29]. However, no studies have examined fatty acid and amino acid contents as quality indicators of high pressure processed clams, particularly New Zealand clams. Therefore, the aim of this study was to determine the amino acids and fatty acids content of three different clams, namely, New Zealand Diamond Shell clams (*Spisula aequilatera*), Storm Shell (*Mactra murchisoni*), and Tua Tua (*Paphies donacina*) using HPP under varying pressures and pressure holding time conditions.

## 2. Materials and Methods

### 2.1. Sample Preparation

New Zealand Diamond Shell (*Spisula aequilatera*), Storm Shell (*Mactra murchisoni*), and Tua Tua (*Paphies donacina*) samples were transported from Cloudy Bay Clams Ltd. located in Cloudy Bay, South Island, New Zealand, by air under chilled conditions. Upon arrival, chilled samples were transported within an hour to the laboratory at Auckland University of Technology, Auckland, New Zealand. All samples were packaged in a high-density polyethylene bag, vacuum packed, and stored frozen at −20 °C two days prior to HPP.

### 2.2. High Pressure Processing

The HPP equipment was a Multivac 55 L soft decompression technology system (MULTIVAC New Zealand Ltd., Auckland, New Zealand) located at FOODBOWL, an open-access facility operated by New Zealand Food Innovation, Auckland, New Zealand. A total of 120 clam samples were randomly separated into 12 packs (10 clams per pack) and subjected to HPP treatment for five s or ten min at 100, 200, 300, 400, 500, and 600 MPa pressure levels at 25 °C. The initial temperature of the water, which was used as the media to transfer pressure, was around 7–8 °C. The temperature of water slightly increased when pressure increased, but no more than 20 °C. All control and samples subjected to HPP treatments were frozen at −20 °C. Twelve vacuum-packed clam samples (each pack containing 10 randomly chosen clam samples) were subjected to HPP treatments at 100, 200, 300, 400, 500, and 600 MPa pressure and held for either 5 s or 600 s. Control samples comprised ten untreated clam samples.

All pressure-treated and control clams were shucked. The meat was freeze-dried for 72 h at room temperature using a Christ Alpha 2–4 LD freeze dryer (Martin Christ Gefriertrocknungsanlagen GmbH, Harz, Germany). Freeze-dried samples were finely milled by using an IKA^®^ A 11 basic analytical mill (IKA^®^-Werke GmbH & Co. KG, Staufen, Germany). Milled clam powder was packed into sealed bags and stored frozen at −20 °C prior to analysis.

### 2.3. Free Amino Acid Analysis

This method involved a solid phase extraction step, derivatization, and finally, a solid/liquid extraction. Clam powder (0.05 g) was weighed as accurately as possible using an analytical balance and placed into a 4 mL centrifuge tube. Methanol (1 mL) was then added to the powder and vortex mixed for 30 min. Samples were centrifuged at 10,000× *g* for 5 min at room temperature. The resulting supernatant (25 μL) was introduced into a sample preparation vial for amino acid extraction using a free amino acid commercial kit, EZ: faast^TM^ (Phenomenex, CA, USA). The internal standard used was norvaline in an n-propanol solution (0.2 mM). Internal standard (100 μL) (0.2 Mm Norvaline + 10% N-propanol) was added into a sample preparation vial. The sample was first passed through a sorbent tip provided. Then, n-propanol (200 μL) was passed through the same tip. Finally, 200 μL of the eluting medium (150 μL sodium hydroxide + 200 μL n-propanol) was passed through the sorbent to elute the sample into the sample preparation vial. Chloroform (50 μL) was added to the sample preparation vial using a micro-dispenser. The matrix was vortexed for 5 s and left standing for one min (repeated two times). Isooctane (100 μL) was added into the sample preparation vial using a micro-dispenser. The matrix was vortexed for 5 s and left standing for one min. We then added 1N Hydrochloric acid (100 μL) into the sample preparation vial. The top organic phase (150 μL) was transferred into a 1.8 mL autosampler vial containing a low-volume glass insert and capped prior to injection into the GC.

A Shimadzu GC2010 gas chromatograph with a Flame Ionisation Detector (GC-FID) (Shimadzu Corporation, Kyoto, Japan), a split/splitless injector, and an AOC-20i auto-injector attached was employed for amino acid analysis. The ZBAAA GC column (10 m × 0.25 mm × 0.25 um, Phenomenex, CA, USA) that was included in the EZ-faast kit was used for amino acid analysis. The instrument was set up according to the EZ-faast kit lab manual. The amino acid samples (10 µL) were injected into the column. The starting temperature of the GC oven was 80 °C. The temperature was then increased to 320 °C using a rate of 25 °C per min and held for a further 3 min at this temperature. Nitrogen with a column flow at 1.46 mL/min was used as carrier gas.

Calibration curves of 19 amino acids (Ser, Gly, b-Iab, Ser, Pro, Asp, Hyp, Glu, Gln, Orn, Tyr, Val, Leu, Ile, Thr, Met, Phe, Lys, and Trp) were constructed following the same procedure for preparation of samples. Each calibration curve was constructed using five different concentrations of individual amino acids adjusted to the expected concentration in the product. The method of internal standard area calibration was followed. The higher concentrations of Ser and Gly in samples necessitated the use of higher standard concentrations (1000 nmol/mL, 2000 nmol/mL, 4000 nmol/mL, and 8000 nmol/mL) for the construction of the standard calibration curve. Amino acid concentrations of clam samples were determined by using the standard curve and the peak area of each amino acid obtained from the chromatogram.

### 2.4. Fatty Acid Analysis

Total fatty acids quantification was carried out [30]. Fatty acids were released by acid hydrolysis of lipids in lyophilized samples. This is followed by in situ esterification of fatty acid methyl esters (FAMEs) and their extraction into toluene prior to GC analysis. The clam powder (20 mg) sample was accurately measured using an analytical balance and put into a 4-mL brown glass vial. Tridecanoic acid (TDA) internal standard solution (10 μL of 2 g/L TDA in toluene) was then added to the vial by using a glass syringe. Toluene (490 μL) and 750 μL of methanolic HCL (5 mL concentrated HCL + 95 mL methanol) were added and vortexed thoroughly to mix. The vials were covered with lids and sealed using a plastic film. The vials were incubated at 70 °C in a heating block for 2 h. After the vials were cooled to room temperature, 6% K_2_CO_3_ (1 mL) and toluene (500 μL) were added and vortexed to mix thoroughly. Then the mixture was centrifuged (1100× *g* for 5 min). The top organic phase was transferred into a 1.8 mL brown autosampler vial containing a low-volume glass insert and cap for injection into the GC.

A Shimadzu GC2010 gas chromatograph fitted with a Flame Ionisation Detector (GC-FID) (Shimadzu Corporation, Kyoto, Japan), a split/splitless injector, and an AOC-20i auto-injector attached was employed for fatty acid analysis. The FAMEWAX (USP G16) column (0.25 mm × 30 m × 0.25 µm) from Restek, USA, was used. The fatty acid samples were automatically injected into the column. The GC conditions were set based on conditions used in a previous study [30]. The starting temperature of the GC oven was 140 °C for the first two min, followed by an increase to 250 °C at a rate of 3.14 °C per min and held for a further 3 min at this temperature. Nitrogen with a column flow at 7 mL/min was used as the carrier gas.

Thirteen calibration curves were constructed for: C10:0, C14:0, C16:0, C18:0, C23:0, C24:0, C16:1, C18:1n9c/C18:1n9t, C20:1n9, C20:3n6, C20:5n3, C22:1n9, and C22:6n3 fatty acids. Each fatty acid curve was made using six concentration levels of the FAME standard adjusted to the expected concentration in the product (Supelco 47885-U, Sigma Aldrich, Sydney, Australia) in 1.5 mL vials. Fatty acids concentration of clam samples were determined by using the standard curve and the peak area of each fatty acid from the chromatogram.

### 2.5. Statistical Analysis

Datasets collected from biological samples are generally known to possess high sample variability. The large variation may cause common problems where subtle small effect size may not be detected due to the increase of noise, and the treatment may differ due to the inherent variation between biological samples. Therefore, Orthogonal Partial Least Squares Discriminant Analysis (OPLS-DA) was first deployed to identify the discriminating amino and fatty acids compounds for all clams that were subjected to HPP. As such, OPLS-DA’s non-predictive variation can be described using orthogonal components, which is useful to explain inherent biological variations or experimental biases. Model parameters such as R^2^ and Q^2^ values were then recorded. All data were centered and reduced prior to OPLSDA analysis. Key amino and fatty acids were identified using Variable of Importance (VIP) values that were extracted from OPLSDA analysis. VIP values above 1.0 are considered to be important to discriminate the clam samples [31].

The amino and fatty acids that were identified to be the key discriminators between samples were further subjected to mixed model Analysis of Variance (ANOVA). Mixed model ANOVA was carried out to understand how varying HPP processing pressures (100 MPa, 200 MPa, 300 MPa, 400 MPa, 500 MPa and 600 MPa) and holding times (5 and 600 s) influenced amino acids and fatty acids content in pressure treated clam samples. When results reached statistical significance at 5% level, mean scores were further separated by pairwise comparison using Fisher’s least significant difference test. In addition, Hotelling-Lawley Multivariate ANOVA (MANOVA), Unidimensional test of means equality, and Canonical Variate Analysis (CVA) were carried out on amino acids, fatty acids, and both amino and fatty acids datasets. The 95% confidence ellipses were plotted to signify product differences on the CVA plots. OPLS-DA was carried out using the MetaboAnalystR 3.2 R package, and CVA and mixed-model ANOVAs were carried out using XLSTAT version 2022.5.1 (Addinsoft, Paris, France).

## 3. Results & Discussion

### 3.1. Identification of Key Differentiating Amino Acids and Fatty Acids

Orthogonal Partial Least Squares Discriminant Analysis (OPLS-DA) was deployed to identify the discriminating amino and fatty acids between samples post-HPP (Figure 1). The two dimensions of OPLS-DA explained 52.5% of the sample variance. Samples were separated between species along t1 (also known as PLS Factor 1), where Diamond Shell clams were loaded with negative scores, and Tua Tua were loaded with positive scores. t2 (also known as PLS Factor 2) partially explained the influence of holding time, where samples within the same pressure are generally loaded higher along the axis. ASP, ILE, LEU, LYS, MET, SER, THR, and VAL were then further identified as discriminating amino acids, while C18:0, C22:1n9, C24:0, and C20:5n3 were identified as discriminating fatty acids. These amino and fatty acids were then subjected to mixed model ANOVA.

### 3.2. Mixed Model ANOVA Analysis for Diamond Shell Clam

The effect of HPP treatments at different pressure levels and holding times for the free amino acids and fatty acids composition of Diamond Shell clams are summarized in Table 1 and Table 2, respectively. It is noted that the aim of this study was to minimally process the clams to ensure that product quality is maintained. Hence, a processing time of 5 s was chosen as being the least processing time, while 5 min (or 600 s) was the ideal processing time for shellfish viral inactivation [32]. The amino acid content was significantly affected (*p* < 0.05) by the different pressure levels. Amino acids significantly increased from the lower to mid-range pressure levels (100–300 MPa) and significantly decreased as the pressure was increased from 400 MPa. The findings agree with a different study on raw beef meat [33,34] where a higher level of free amino acids was observed after HPP was applied between 100 and 300 MPa (10 min at 25 °C). Similarly, HPP was found to increase the total free amino acid composition significantly compared to control at all pressures for different lamb cuts [35]. The decrease in amino acid concentration following 400 MPa could be due to a slower time of solubilization from muscle proteins compared to their transformation into biogenic amines through decarboxylation [36]. Furthermore, muscle proteins are also vulnerable to oxidative reactions that could result in a decrease in protein digestibility [37]. A similar decreasing trend in free amino acids concentration at higher pressure levels (>400 MPa) was also observed in another study on freeze-dried squid muscle samples [26]. The authors postulated that at lower pressure levels, amino acid concentrations increased due to improved proteolysis by active enzymes. A decrease in the concentration of certain amino acids, on the other hand, can be caused by the activation of certain amino acid metabolism pathways.

The influence of different holding times at the same pressure on amino acid content showed significance after HPP treatment of Diamond Shell clams for ASP, LEU, and SAR. The interaction effect of pressure and holding times on amino acid content were all significant except for LYS.

Mixed model ANOVA results showed that the FAs composition for Diamond Shell clams was significantly affected (*p* < 0.05) by the different pressure levels and their interaction with processing times, showing a significant increasing trend. This finding is supported by a study [37] that found the total SFA in Yak (*Poephagus grunniens*) body fat after pressure treatment at 600 MPa significantly increased by 10% compared to the control. Results showed a significant decrease in C10:0 after HPP treatment of oysters (*Crassostrea gigas*) at 260 MPa for 3 min and 500 MPa for 5 min by 89% and 82%, respectively, compared to control [38]. Similarly, another study [37] also found that C10:0 significantly decreased at 600 MPa by 28% compared to the control. In a more recent article, the amount of SFA was significantly higher in lamb samples that were HPP-treated [34]. The influence of different holding times was only significant for C18:0 fatty acid in Diamond Shell clams, where it was significantly the highest when the pressure level was 200 MPa.

### 3.3. Canonical Variate Analysis for Diamond Shell Clam

CVA plots indicated that amino acids (Figure 2a) discriminated the samples better compared to fatty acids (Figure 2b). When taken holistically (Figure 2c), some interesting observations emerged, where samples processed with lower pressures (100–300 MPa) were loaded with negative scores, while samples processed at higher pressures (>300 MPa) had positive scores along Factor 1(F1). All samples except 200 MPa showed a tendency to have positive scores along Factor 2 (F2) with longer holding time. Additionally, some samples processed with lower pressures and longer holding time were not significantly different with some samples at high pressure and shorter holding time (e.g., 200 MPa at 600 s with 400 MPa at 5 s). This is shown in Figure 2c when the 95% confidence ellipses overlap. Amino acids and fatty acids that were previously selected through OPLS-DA reached statistical significance in the unidimensional test of means equality, further validating that the use of a fingerprinting technique prior to ANOVA was advantageous in identifying chemical quality indicators.

**Table 1 foods-12-01162-t001:** Free amino acid composition of Diamond Shell (*Spisula aequilatera*) clams. a,b means with different letters in a row show the significant effect of time in each processing pressure; v,w,x,y,z means with different letters in column show the significant effect of time under the same processing pressure. * Compounds identified with VID coefficients higher than 0.8.

Time (s)	Amino Acid	Pressure (MPa)						Significance		
		100	200	300	400	500	600	Processing	Time	Processing * Time
5	ASP	333.48 aw	417.83 aw	832.57 by	533.8 bx	170.02 bx	180.79 av	<0.0001	<0.0001	<0.0001
600		588.67 bx	58.41 ax	205.44 avw	253.13 aw	163.27 aw	171.24 av			
5	ILE	92.34 awx	172.3 av	113.93 ay	107.64 ay	95.82 ay	83.3 bw	<0.0001	0.155	<0.0001
600		78.99 aw	447.13 ay	132.98 by	113.66 axy	92.96 axy	54.91 av			
5	LEU	147.78 bwx	108.59 av	169.72 ay	156.66 axy	145.03 axy	135.29 bw	<0.0001	0.002	<0.0001
600		123.48 av	62.3 aw	194.97 bw	186.53 bw	170.39 bw	108.25 av			
5	LYS	85.98 awx	114.19 bvw	94.69 axy	73.98 avwx	64.1 avwx	64.56 bv	<0.0001	0.947	0.068
600		92.89 ax	163.37 bx	80.59 ax	77.51 awx	68.97 awx	48.92 av			
5	MET	45.56 bx	90.68 av	47.28 axy	41.41 awx	38.31 awx	34.64 bv	<0.0001	0.165	0.001
600		34.08 avw	31.46 av	52.51 av	44.25 axy	42.03 axy	26 av			
5	SER	2365.56 aw	2272.99 aw	2816.26 bxy	2477.28 bwx	1992.72 bwx	1884.61 av	<0.0001	<0.0001	<0.0001
600		2415.51 aw	2094.06 av	2042.82 av	1946.59 av	1869.59 av	1884.1 av			
5	THR	79.72 av	101.75 av	170.45 axy	184.05 ax	161.67 ax	136.1 bw	<0.0001	0.108	<0.0001
600		88.47 av	174.08 bwx	189.18 bx	186.85 bx	151.51 bx	94.34 bv			
5	VAL	143.12 bx	111.43 av	168.62 ay	168.28 ay	147.64 ay	126.42 bw	<0.0001	0.087	<0.0001
600		122.56 aw	170.47 bxy	198.86 by	173.14 axy	147.37 axy	92.73 av			

**Table 2 foods-12-01162-t002:** Fatty acids composition of Diamond Shell (*Spisula aequilatera*) clam. a,b means with different letters in a row show the significant effect of time in each processing pressure; v,w,x,y,z means with different letters in column show the significant effect of time under the same processing pressure. * Compounds identified with VID coefficients higher than 0.8.

Time (s)	Fatty Acid	Pressure (MPa)						Significance		
		100	200	300	400	500	600	Processing	Time	Processing * Time
5	C18:0	6 avw	0.66 bw	5.91 av	5.84 av	6.41 aw	6.13 avw	<0.0001	0.012	0.001
600		6.27 ax	6.08 avw	5.77 avw	6.45 bwx	6.84 bv	6.55 bvw			
5	C20:5n3	5.67 av	6.6 bw	5.94 avw	6.19 avw	6.28 avw	6.14 avw	0	0.554	0
600		6.08 ax	5.58 ax	5.4 ax	6.62 awx	6.92 aw	6.63 av			
5	C22:1n9	1.51 aw	1.86 bx	1.5 aw	1.33 av	1.71 bx	1.52 bw	<0.0001	0.145	<0.0001
600		1.62 avw	1.56 av	1.49 av	1.62 bxy	1.52 awx	1.38 av			
5	C24:0	2.05 avwx	2.26 bwx	1.8 av	1.97 avw	2.37 ax	1.91 avw	0.006	0.119	0.007
600		1.84 ay	1.79 az	1.77 az	2.31 ax	2.04 aw	2.02 av			

**Figure 2 foods-12-01162-f002:**
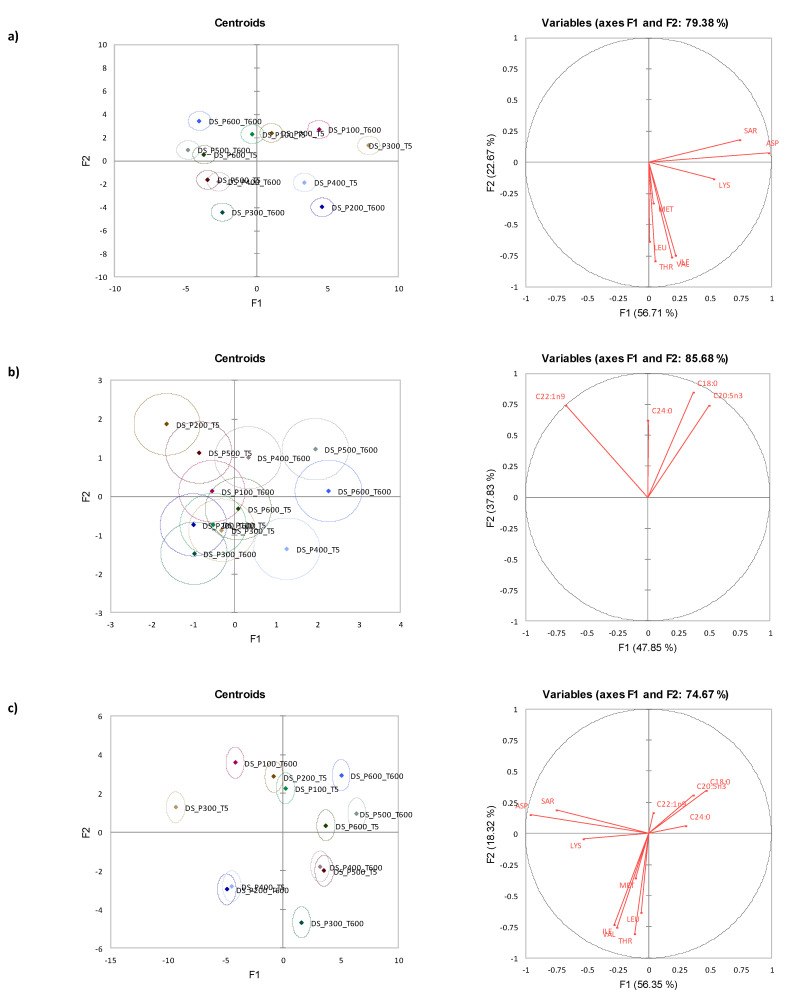
Canonical variate analysis centroids (left) and vectors (right) for (**a**) free amino acid (F_(88,410)_ = 19.917, *p* < 0.001), (**b**) fatty acids (F_(44,222)_ = 4.385, *p* < 0.001), and (**c**) both free amino acids and fatty acids (F_(132,233)_ = 16.230, *p* < 0.001) of Diamond Shell (*Spisula aequilatera*) clam.

### 3.4. Mixed Model ANOVA Analysis for Storm Shell Clam

The effect of HPP treatments at different pressure levels and holding times on the free amino acids and fatty acids composition of Storm Shell clam are summarized in Table 3 and Table 4, respectively. The amino acid content of Storm shell clam was significantly affected (*p* < 0.05) by the different pressure levels. Similar to the Diamond Shell clam, the amino acids significantly increased from the lower to mid-range pressure levels (100–300 MPa), and significantly decreased as the pressure was increased from 400 MPa. In contrast to the Diamond Shell clam, the influence of holding time was significant for all amino acids except one (MET) in Storm Shell clams. Amino acids in Storm Shell clams were significantly affected by the pressure and holding times interactions.

The influence of pressure levels on the fatty acid compositions of Storm Shell clams was significant for all fatty acids except for C20:5n3 fatty acid. The effect of holding time and its interactions (processing and time) significantly influenced almost all fatty acids except for C24.0 fatty acid.

### 3.5. Canonical Variate Analysis for Storm Shell Clams

Similar to the Diamond Shell clam results, CVA plots indicated that amino acids (Figure 3a) discriminated the HPP processed samples better compared to fatty acids (Figure 3b). Figure 3c showed that the merged results for both amino and fatty acids, further enhanced discrimination between samples with HPP processing in terms of both fatty acid and amino acid data.

CVA showed that there was no effect of holding time under some pressure conditions (i.e., 200 MPa, 400 MPa, and 500 MPa). There was no clear separation of samples for pressure and holding time along F1 and F2. All amino acids and fatty acids that were previously selected through OPLS-DA reached statistical significance in the unidimensional test of means equality, further validating the fingerprinting technique that was carried out to identify chemical quality indicators when clams were subjected to HPP processing.

### 3.6. Mixed Model ANOVA Analysis for Tua Tua

The effect of HPP treatments at different pressure levels and holding times for the free amino acids and fatty acids composition of Tua Tua are summarized in Table 5 and Table 6, respectively. The levels of amino acid content for Tua Tua were significantly affected (*p* < 0.05) by the different pressure levels. The changes in amino acid composition for Tua Tua after HPP treatments were slightly different compared to Storm and Diamond shell clams in that most of the amino acid content significantly decreased as the pressure level increased. For example, the amino acids like MET, SAR, and VAL decreased as the pressure level increased. The other amino acids significantly increased slightly after 200 MPa, and then significantly decreased as the pressure level increased from 400 to 600 MPa. It is interesting to observe the different trends in amino acid composition when the clams were subjected to different pressure levels. The decomposition of amino acids can vary from each other with respect to the hydrophobicity index of their side chains. In a study investigating the reaction kinetic behavior during the decomposition of five amino acids in high-pressure water, it was found that the decomposition rates of hydrophilic amino acids tended to be higher than the hydrophobic amino acids [39]. In this case, it may be possible that the three amino acids in the current research, namely MET, SAR, and VAL, that had a lower hydrophobic index compared to other types of amino acids decreased in concentration as they formed degradation products.

The influence of holding time affected almost all the amino acids in HPP-treated Tua Tua clams significantly, except for SAR. The interactions of pressure and holding time also significantly affected all amino acids.

The fatty acids were significantly influenced by the pressure levels. However, there were no obvious increasing or decreasing trends observed. The effect of pressure level was only not significant for C20:5n3 fatty acid. The influence of holding time was only significant for C18.0 fatty acid, and the interactions (processing and time) were only significant for C24.0 fatty acid. The small changes in fatty acids with increasing pressure (significantly small decrease between 100 to 300 MPa and increase after 400 MPa) (*p* < 0.05) were not surprising. Similar findings where there were no obvious increasing and decreasing trend in fatty acids content as pressure levels increased have been reported for other species such as Korean native black goat meat [40], salmon and oysters [23,38] and beef [41]. 

### 3.7. Canonical Variate Analysis for Tua Tua Clams

CVA plots indicated that selected amino acids (Figure 4a) discriminated the the HPP processed Tua Tua clam samples better compared to fatty acids (Figure 4b). Figure 4c shows the merged results for both amino and fatty acids where similarly, the contribution of amino acids and fatty acids assisted in the discrimination of the HPP sample evidently with smaller confidence ellipses in the product. There was no clear pattern that explained the effects of pressure and holding time along F1 and F2. However, there are some interesting samples to note. Tua Tua clam samples subjected to 400 MPa and 500 MPa treatments were not significantly different, with shorter holding times. Additionally, samples treated at 500 MPa that was subjected to a long holding time were not significantly different compared to sample treated at 600 MPa held for a short time. 

## 4. Conclusions

This study sets out to explore the influence of different HPP pressure levels and holding times on the amino acids and fatty acids content of three different New Zealand clams. The use of multivariate analysis such as OPLS-DA revealed that the discriminating amino acids between samples post-HPP processing were ASP, ILE, LEU, LYS, MET, SER, THR, and VAL, while C18:0, C22:1n9, C24:0, and C25:5n3 were identified as discriminating fatty acids. Significant effects of pressure levels were found following a mixed model ANOVA analysis. Amino acids significantly increased from the lower to mid-range pressure levels (100–300 MPa) and decreased as pressure was increased from 400 MPa. The influence of pressure levels on fatty acids was significant for all three types of New Zealand clams. A longer holding time, particularly between the low and mid-range of pressure levels (100–400 MPa), significantly decreased the free amino acids content of all three types of clams. The holding time significantly decreased only a few types of fatty acids. This suggests that HPP hardly caused hydrolysis of triglycerides.

It can be concluded that lower pressure levels and short pressure holding time, when applied to clams, maintained amino acids and fatty acids content. Results from our study showed that HPP has great potential in processing clams. However, prolonged pressure and time can increase the temperature of clams, which may contribute to the further denaturation of protein. It is noted that in this study, all variables (e.g., HPP processing parameters and species variation) were subjected to OPLS-DA to identify the key discriminating amino and fatty acids holistically for all three clam species. If one is interested in discounting species variation, separate OPLS-DA can be attempted per species to identify the key amino acids and fatty acids that discriminated the samples. Further research to investigate how HPP processing affects volatile flavor compounds, texture and rancidity of clams with chilled or frozen storage is necessary either using static [42] or temporal sensory technique [43] (e.g., TCATA) can be carried out. The use of a wider and more comprehensive experimental design is necessary before HPP can be applied to further develop the next generation of HPP seafood products in the market.

## Figures and Tables

**Figure 1 foods-12-01162-f001:**
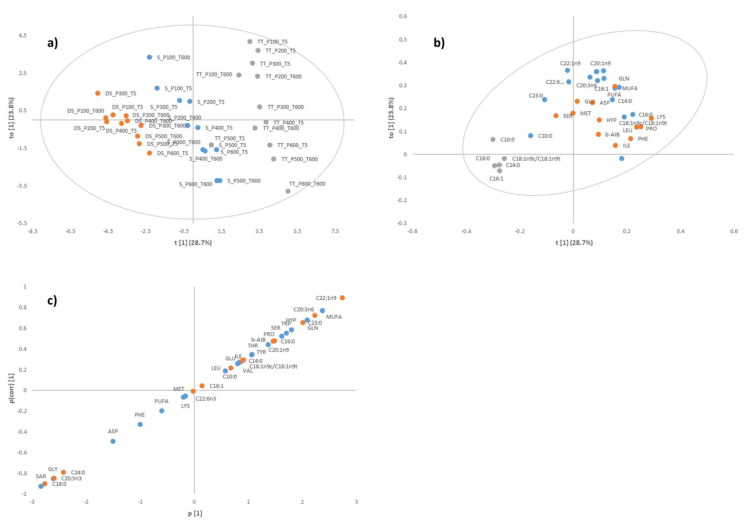
Orthogonal Partial Least Squares Discriminant Analysis (OPLS-DA) for all samples based on fatty acids and amino acid measures; (**a**) sample score plot (orange: Diamond Shell, blue: Storm, grey: Tua Tua), (**b**) loading plot (orange: fatty acid, blue: amino acid, grey: discriminant fatty acid and amino acid), (**c**) S-plot (orange: fatty acid, blue: amino acid). Assessed model parameters-R^2^X = 0.287, R^2^Y = 0.787 and Q^2^ coefficient = 0.785.

**Figure 3 foods-12-01162-f003:**
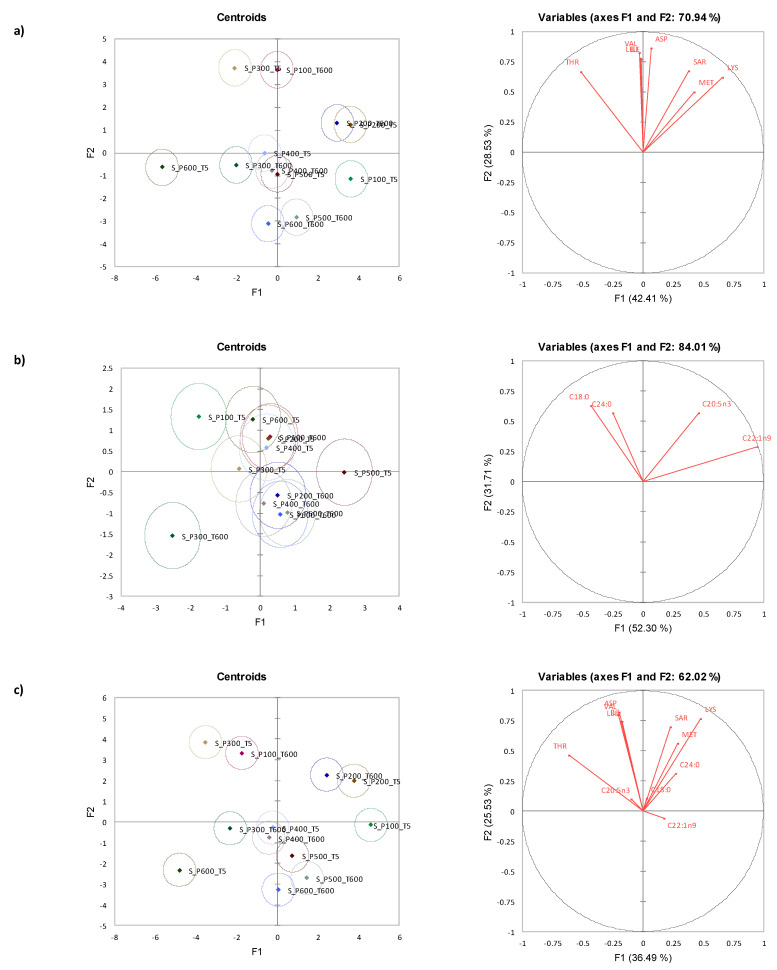
Canonical variate analysis centroids (left) and vectors (right) for: (**a**) free amino acids (F_(88,410)_ = 10.515, *p* < 0.001), (**b**) fatty acids (F_(44,222)_ = 4.124, *p* < 0.001), and (**c**) both free amino acids and fatty acids (F_(132,233)_ = 8.666, *p* < 0.001) of Storm Shell (*Mactra murchisoni*) clam.

**Figure 4 foods-12-01162-f004:**
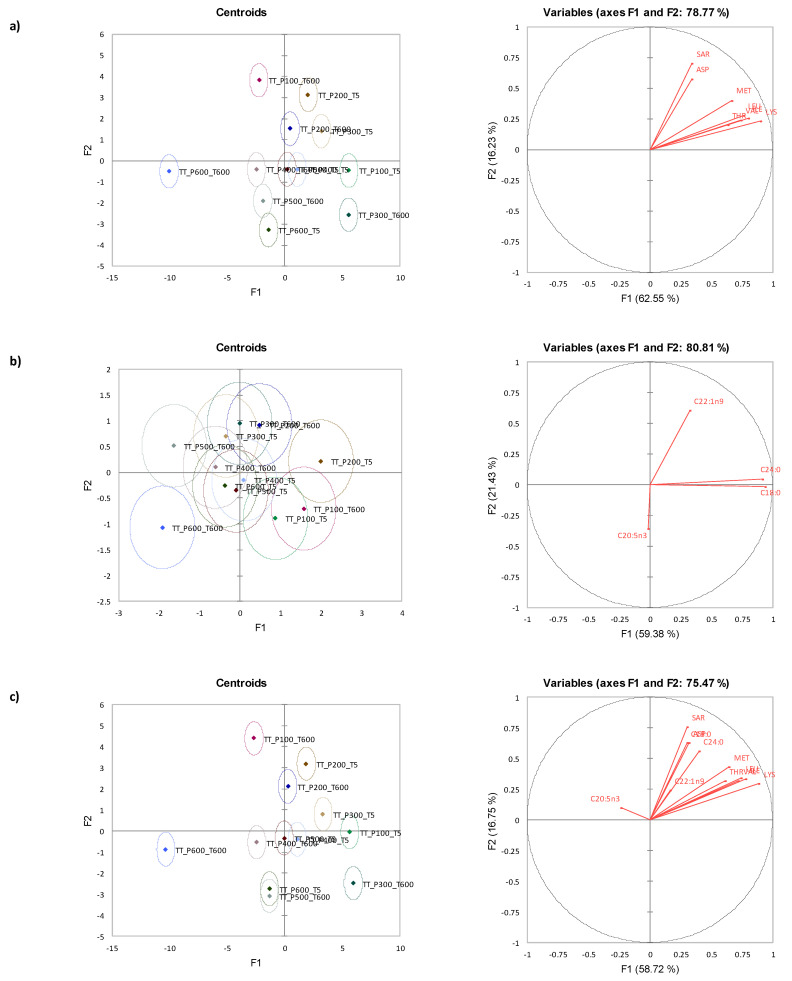
Canonical variate analysis centroids (left) and vectors (right) for (**a**) free amino acids (F_(88,410)_ = 18.129, *p* < 0.001), (**b**) fatty acids (F_(44,222)_ = 3.069, *p* < 0.001), and (**c**) both free amino acids and fatty acids (F_(132,233)_ = 12.650, *p* < 0.001) of Tua Tua (*Paphies donacina*) clam.

**Table 3 foods-12-01162-t003:** Free amino acid composition of Storm Shell (*Mactra murchisoni*) clam. a,b means with different letters within a row show the significant effect of time at each processing pressure; v,w,x,y,z means with different letters within column show the significant effect of time under the same processing pressure. * Compounds identified with VID coefficients higher than 0.8.

Time (s)	Amino Acid	Pressure (MPa)						Significance		
		100	200	300	400	500	600	Processing	Time	Processing * Time
5	ASP	355.35 av	546.06 ax	787.6 by	469.1 awx	396.88 avw	406.27 avw	<0.0001	0.024	<0.0001
600		839.98 by	679.18 ax	368.76 av	509.19 bw	418.86 avw	385.85 av			
5	ILE	65.57 av	96.52 awx	112.5 ax	79.26 bvw	70.01 bv	71.67 bv	<0.0001	0	0.001
600		88.15 bx	86.69 ax	81.63 ax	64.24 aw	44.18 av	40.5 av			
5	LEU	92.63 av	126.47 awx	130.51 ax	98.69 bvw	88.15 bv	88.81 bv	<0.0001	<0.0001	<0.0001
600		125.49 bx	91.54 aw	120.23 ax	77.66 aw	41.47 av	42.7 av			
5	LYS	65.29 ax	79.19 ay	52.18 awx	45.32 aw	39.92 aw	18.91 av	<0.0001	0.022	<0.0001
600		72.31 az	61.35 ay	46.04 ax	42.11 awx	33.45 avw	26.3 bv			
5	MET	23.11 awx	30.25 bx	15.19 avw	15.78 bvw	10.15 av	10.51 av	<0.0001	0.24	<0.0001
600		32.22 ax	15.43 aw	13.92 avw	14.77 aw	11.89 avw	10.67 av			
5	SAR	1729.13 aw	1601.77 aw	1684.01 bw	1511.8 bw	1586.72 bw	1262.01 av	<0.0001	<0.0001	0
600		1847.52 ay	1626.16 ax	1238.17 avw	1356.17 aw	1213.05 avw	1160.61 av			
5	THR	154.44 avw	135.23 av	217.12 bx	173.74 bw	159.31 bvw	218.57 bx	<0.0001	<0.0001	<0.0001
600		226.48 bx	137.22 aw	162.62 awx	143.83 aw	101.62 av	107.17 av			
5	VAL	104.18 av	154.55 awx	171.83 ax	123.07 bvw	105.87 bv	113.94 bv	<0.0001	<0.0001	<0.0001
600		138.73 bx	134.88 ax	143.47 ax	100.22 aw	62.32 av	56.7 av			

**Table 4 foods-12-01162-t004:** Fatty acid composition of Storm Shell (*Mactra murchisoni*) clam. a,b means with different letters within a row show the significant effect of time for each processing pressure; v,w,x,y,z means with different letters within a column shows the significant effect of time under the same processing pressure. * Compounds identified with VID coefficients higher than 0.8.

Time (s)	Fatty Acid	Pressure (MPa)						Significance		
		100	200	300	400	500	600	Processing	Time	Processing * Time
5	C18:0	4.66 bx	3.75 avwx	3.09 avwx	3.59 bvwx	2.59 av	4.3 awx	0.014	<0.0001	0.017
600		2.99 avw	2.84 avw	3.36 aw	2.65 avw	2.32 av	2.23 av			
5	C20:5n3	1.26 av	1.45 av	1.41 bv	1.43 bv	1.47 av	1.55 av	0.118	0.019	0.012
600		1.58 bx	1.28 aw	0.94 av	1.3 aw	1.32 aw	1.33 aw			
5	C22:1n9	3.15 av	3.69 av	3.33 bv	3.65 bv	4.41 aw	3.5 av	<0.0001	0.028	<0.0001
600		3.79 bx	3.66 awx	2.44 av	3.47 awx	3.68 awx	3.61 awx			
5	C24:0	0.83 aw	0.58 av	0.57 bv	0.57 bv	0.53 av	0.47 av	0	0.028	0.923
600		0.68 aw	0.53 av	0.48 av	0.49 av	0.45 av	0.45 av			

**Table 5 foods-12-01162-t005:** Free amino acid composition of Tua Tua (*Paphies donacina*) clam. a,b means with different letters within a row show the significant effect of time with each processing pressure; v,w,x,y,z means with different letters within column show the significant effect of time under the same processing pressure. * Compounds identified with VID coefficients higher than 0.8.

Time (s)	Amino Acid	Pressure (MPa)						Significance		
		100	200	300	400	500	600	Processing	Time	Processing * Time
5	ASP	193.12 avw	212.39 w	175.24 avw	172.33 av	168.57 av	157.34 av	<0.0001	0.104	0.009
600		245.88 bz	215.51 ay	196.72 axy	181.82 bwx	163.11 aw	132.89 av			
5	ILE	173.82 bx	157.6 wx	168.53 ax	136.42 bw	131.02 bvw	103.84 bv	<0.0001	<0.0001	<0.0001
600		128.89 ax	183.55 ay	192.82 ay	116.83 ax	77.21 aw	54.36 av			
5	LEU	255.47 by	224.56 wxy	240.17 axy	198.03 bvwx	180.92 bvw	157.04 bv	<0.0001	<0.0001	<0.0001
600		207.39 awx	230.35 axy	264.07 ay	172.97 aw	128.77 av	102.27 av			
5	LYS	127.95 by	98.53 wx	108.31 ax	89.68 bw	84.81 bvw	68.61 bv	<0.0001	<0.0001	<0.0001
600		92.11 ax	101.78 axy	115.14 ay	68.59 aw	64.01 aw	9.71 av			
5	MET	65.12 ax	58.69 x	55.23 ax	40.23 bw	33.17 aw	16.7 av	<0.0001	0.032	<0.0001
600		59.62 ayz	43.46 axy	69.77 az	36.85 awx	24.35 avw	14.82 av			
5	SAR	1109.24 aw	1148.45 w	1161.47 bw	1012.66 avw	1038.09 bvw	862.02 av	<0.0001	0.226	<0.0001
600		1287.37 bx	1378.6 ay	964.09 aw	988.13 aw	770.51 av	796.04 av			
5	THR	270.91 bw	206.95 v	240.52 avw	207.88 av	197.4 bv	197.85 bv	<0.0001	0.072	<0.0001
600		236.16 aw	303.21 bx	295.73 ax	196.32 aw	109.25 av	111.43 av			
5	VAL	254.51 bx	230.57 wx	244.92 awx	204.62 bvwx	203.2 bvw	162.52 bv	<0.0001	0.001	<0.0001
600		199.35 aw	288.37 ax	286.16 ax	184.49 aw	127.27 av	93.92 av			

**Table 6 foods-12-01162-t006:** Fatty acid composition of Tua Tua (*Paphies donacina*) clam. a,b means with different letters within a row show the significant effect of time at each processing pressure; v,w,x,y,z means with different letters within column show the significant effect of time under the same processing pressure. * Compounds identified with VID coefficients higher than 0.8.

Time (s)	Fatty Acid	Pressure (MPa)						Significance		
		100	200	300	400	500	600	Processing	Time	Processing * Time
5	C18:0	1.82 awx	2.13 ax	1.45 av	1.6 bvw	1.46 av	1.53 bvw	<0.0001	0.024	0.103
600		1.95 ax	1.82 ax	1.51 aw	1.47 avw	1.26 avw	1.21 av			
5	C20:5 n3	0.96 aw	0.91 avw	0.82 av	0.89 avw	0.87 avw	0.91 avw	0.163	0.223	0.871
600		0.93 av	0.98 av	0.85 av	0.93 bv	0.88 av	1.01 av			
5	C22:1n9	7.11 avw	8.06 aw	7.14 avw	7.07 avw	6.5 av	7.05 avw	0.005	0.239	0.748
600		7.19 av	8.67 aw	7.48 av	7.34 bv	7.26 av	6.65 av			
5	C24:0	0.7 avw	0.79 aw	0.6 av	0.63 bvw	0.64 bvw	0.58 bv	<0.0001	0.006	0.004
600		0.77 ay	0.67 ax	0.66 awx	0.57 aw	0.46 av	0.44 av			

## Data Availability

The data are available from the corresponding author.

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
