# Peer review of "Using OPLS-DA to Fingerprint Key Free Amino and Fatty Acids in Understanding the Influence of High Pressure Processing in New Zealand Clams"

_foods, 2023, doi:10.3390/foods12061162_

Round 1
Reviewer 1 Report
This study investigates the effect of HPP pressures and holding time on free amino acids and free fatty acids of clams by multivariate analysis. It is an interesting topic, and it is useful for extending the shelf life and for improving the eating quality of clams. However, there are still spaces for improvement. Therefore, I suggest the authors to major revise this manuscript.
Regrading to the general comments, there are three aspects.
1. The structure of introduction is not clear. Readers cannot capture the innovative points of this study. Could you explain why you choose FFA and FAA as the indicators for evaluating HPP, and your research gap and objectives are not novel. Please rephase this section.
2. The sampling collection is confusing. The authors didn’t explain why they chose different species and didn’t introduce the differences of three clams. However, in the discussion section, the authors discussed them separately. Any discriminative results from OPLS-DA might be due to the sampling variances. Therefore, the discussions are too premature.
3. In the results and discussions section, I suggest adding more discussion about why different pressures would influence FFA/FAA, or why different FFA/FAA would be distinctive from each other from the protein denaturation, molecule structure aspects etc.
More comments and suggestions are marked in the manuscript in yellow.

Author Response
We'd like to thank the reviewers for the comments. Detailed response can be found attached.

Reviewer 2 Report
The scientific value of the manuscript is high and the subject investigated is interesting to read. The methodology employed is appropriate and the results have been presented in an understandable way. The only major question about the experimental design of the manuscript is that why the authors chose 5 s and 600 s for processing time. There is a extremly wide range between these two processing times and one is expect that with the increase in processing time the impact of HHP on the test subject would be more visible. Is 5 s processing time a common practice in clams processing? Addiitonally, it is known that with the increase in pressure and holding time the temperature of the test/process materials increase (up to 20 C). Here 20 C processing temperature is expected to increase during processing, being more pronounced at higher pressures and longer holding times. Did the author make an observation on how this temperature increase might affect the amino acids and fatty acids profiles of the samples?
Author Response

(The authors gave the same response as above.)
